# Predicting Empirical AI Research Outcomes with Language Models

**Jiaxin Wen[1], Chenglei Si[2], Chen Yueh-Han[3], He He[3], Shi Feng[4]**
[1]UC Berkeley [2]Stanford [3]New York University [4]George Washington University

## Abstract

Many promising-looking ideas in AI research fail to deliver, but their validation takes substantial human labor and compute. Predicting an idea's chance of success is thus crucial for accelerating empirical AI research, a skill that even expert researchers can only acquire through substantial experience. We build the first benchmark for this task and compare LMs with human experts. Concretely, given two research ideas (e.g., two jailbreaking methods), we aim to predict which will perform better on a set of benchmarks. We scrape ideas and experimental results from conference papers, yielding 1,444 human-verified idea pairs *published after our base model's cut-off date* for testing, and 6,000 pairs for training. We then develop a system that combines a fine-tuned GPT-4.1 with a paper retrieval agent, and we recruit 25 human experts to compare with. In the NLP domain, our system beats human experts by a large margin (64.4% v.s. 48.9%). On the full test set, our system achieves 77% accuracy, while off-the-shelf frontier LMs like o3 perform no better than random guessing, even with the same retrieval augmentation. We verify that our system does not exploit superficial features like idea complexity through extensive human-written and LM-designed robustness tests. Finally, we evaluate our system on unpublished novel ideas, including ideas generated by an AI ideation agent. Our system achieves 63.6% accuracy, demonstrating its potential as a reward model for improving idea generation models. Altogether, our results outline a promising new direction for LMs to accelerate empirical AI research.

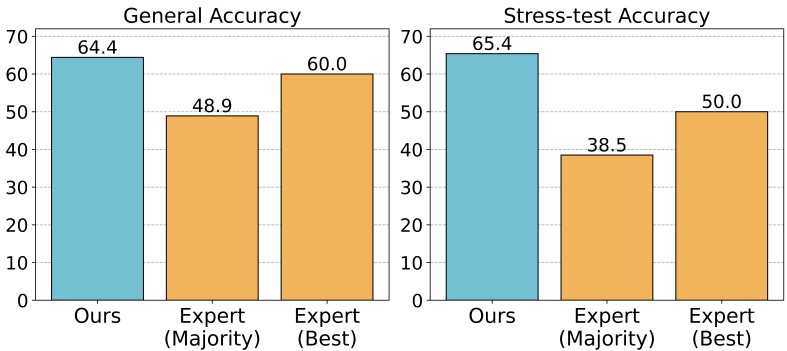

Figure 1: **Our system is more accurate than human NLP experts**. *Majority* aggregates predictions from all annotators, while *Best* keeps only the best-performing annotator per research topic. For the stress test, we select a subset of idea pairs where the mathematically complex one is actually ineffective. While humans often get misled by this feature, our model does not.

39th Conference on Neural Information Processing Systems (NeurIPS 2025).

# 1 Introduction

Many promising-looking AI research ideas turn out to be ineffective when executed. Unfortunately, the only way to find out is to actually implement them. The failed ideas can add up to a significant cost in both human labor and computational resources. Better prioritization—implementing the more promising ideas first—can thus significantly improve research efficiency. But doing so requires predicting the outcomes of experiments without actually running them, a seemingly impossible task.

We hypothesize that language models (LMs) can do this task better than human experts. Humans develop such research intuition through experience, but LMs can acquire it more efficiently by consuming countless research papers, analyzing experimental results, and potentially discovering subtle patterns that are difficult for humans to identify.

Concretely, given the description of a pair of research ideas, our goal is to predict which one works better on a set of benchmarks. For example, given two jailbreaking methods, we aim to predict which one achieves higher attack success rates. This is an objective task: we can obtain the groundtruth by actually implementing both ideas. But making a prediction without implementing the ideas is difficult, and any non-trivial accuracy could be valuable as it informs resource allocation, prioritization of experiments, and iterative refinement of ideas. Crucially, this task provides more actionable information than evaluating subjective aspects of ideas such as novelty or excitement [14, 9, 4], which existing work focuses on.

We construct a benchmark to facilitate the study of this task by scraping both ideas and results from existing conference papers. Each example consists of a research goal specified by a set of benchmarks, two competing ideas, and a binary outcome label indicating which idea performs better across these benchmarks. After four rounds of human verification, we obtain 1,444 verified test examples. To avoid data contamination, we ensure that each test example contains at least one idea published after July 1st, 2024, the knowledge cut-off date of our base model GPT-4.1.

We then develop a system that combines a specialized GPT-4.1 fine-tuned on 6,000 historical idea pairs from the training set and a paper retrieval agent. This system achieves a promising 77% accuracy on the test set. Off-the-shelf frontier models (e.g., o3, Claude 3.5 Sonnet)—even when augmented by the same paper retrieval agent—perform no better than random guessing, demonstrating the importance of proper capability elicitation. We then identify a challenging test subset consisting of 45 NLP idea pairs, and recruit 25 NLP experts to establish a human baseline. Specifically, each human prediction is made by an ensemble of 5 experts who collectively spent over 45 minutes. Our system beats this strong expert baseline by a large margin (64.4% v.s. 48.9%).

To test the system's generalizability, we design stress tests to measure its sensitivity to superficial features like idea recency and complexity. On three human-designed stress tests and hundreds tests proposed by LMs, our system demonstrates robust behavior.

Lastly, we test our system on brand new, unpublished research projects from a recent study on using AIs to generate ideas [14]. This is a set of 35 ideas with experiments implemented by NLP researchers (recruited by the study's authors). These projects have never been released publicly anywhere on the internet, thus avoiding any possible contamination. Our system achieves an accuracy of 63.6%, indicating its generalizability and its potential to improve existing automated research systems [14, 9] as an idea ranker or a reward model.

# 2 Benchmark

We define our task as predicting the outcomes of two competing research ideas for a given research goal. We focus on pairwise evaluation, where the binary outcome label is aggregated over multiple benchmarks to avoid the ambiguity and noise when evaluating individual ideas [14, 9, 4]. Each example involves the following components (Figure 2):

- **Idea pair**, each defined by a detailed description following a standard format.
- **Empirical research goal**, defined by a set of benchmarks, each with a quantitative metric.
- **Binary outcome label**, indicating which idea wins on more benchmarks.

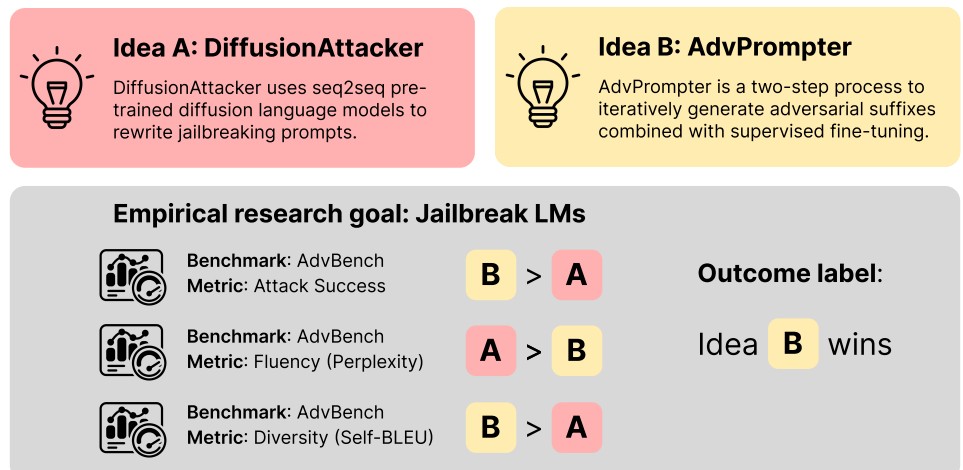

Figure 2: Comparing two ideas for jailbreaking methods on three benchmarks. Each idea is defined by a detailed summary. The goal is to predict the outcome, whose label is determined by actually evaluating the ideas on these benchmarks and seeing who wins more.

## 2.1 Benchmark Construction

We develop an automatic pipeline to extract ideas and their corresponding empirical results from existing papers. The pipeline operates in four steps using Claude 3.5 Sonnet as the main LM.

**Step 1: Paper Collection.** We start by collecting papers from top AI conferences in various domains. Table 1 outlines our data sources. To collect test examples published after frontier LMs' knowledge cut-off, we mainly collect papers published in the past two years. After scraping, we use LMs to filter out non-empirical papers, such as those focus on "rethinking", and "understanding" topics.

**Step 2: Data Extraction.** For each paper PDF, we prompt LMs to extract the research goal, idea names and their outcomes, mainly from the result tables. Next, we generate summaries for each idea based on the PDF. While the main idea is explained in detail, baseline ideas are often described only briefly and potentially in a biased way. To address this, we extract the reference for each baseline idea and download the corresponding paper PDF. If we fail to find references (e.g., the reference is missing or the extracted reference is invalid), we discard these ideas. To avoid information leakage, we explicitly instruct LMs not to mention any actual empirical results in summaries.

**Step 3: Idea Pairing.** We then group ideas into comparison pairs. Pairs are created only within the same paper, since cross-paper comparisons might suffer from confounding factors (e.g., differences in hyperparameters or data preprocessing). For example, comparing the results of two parameter-efficient tuning methods reported in different papers (e.g., LoRA and prompt tuning) could be invalid if they use different numbers of trainable parameters.

**Step 4: Label Aggregation.** To determine the final comparison label for each idea pair, we do majority voting over all benchmarks. For example, if Idea A outperforms Idea B on 4 out of 5 benchmarks, we label the pair as "Idea A is better". This aggregation strategy mitigates label noise in individual benchmark comparisons. To further improve label quality, we remove idea pairs that result in a tie or are evaluated on fewer than three benchmarks.

**Train/Test Splitting.** We then split our collected data into train and test sets using our main base model GPT-4.1's cut-off date, July 1st, 2024. In particualr, we use the date that the paper first appeared on arxiv (which we extract from its arxiv ID), not the date that it is published at the conference. This ensures no leakage from the fine-tuning process. So the model must predict ideas proposed in the "future work". Data statistics are shown in Table 2.

## 2.2 Human Verification of Extracted Examples

Since our benchmark is automatically constructed by parsing paper PDFs with LMs, parsing errors are inevitable. To ensure the quality and reliability of our test set, we recruit human annotators to

| Table 1: Data source. | |
|---|---|
| **Domain** | **Conferences** |
| **NLP** | ACL, EMNLP, NAACL, COLM |
| **ML** | NeurIPS, ICLR |
| **CV** | CVPR, ECCV, ACMMM |
| **Robotics** | CoRL |

| Table 2: Data statistics. | | |
|---|---|---|
| **Split** | **Train** | **Test** |
| **Size** | 6,000 | 1,444 |
| **# Benchmark** | 3.2 | 3.4 |
| **After Cut-off** | No | Yes |
| **Human-Verified** | No | Yes |

carefully go through our test set, rewriting or removing problematic examples. Note that this set of human annotators are tasked only to verify the extracted examples; the annotators for our expert baseline have much higher expertise in their respctive domains.

**Recruiting Human Annotators.** We select 16 college students majoring in Computer Science or Electronic Engineering out of 30 pre-screened candidates. All of them are experienced in reading AI papers, and 25% of them have published papers at AI conferences. We train annotators to do the verification task with 50 warmup examples and verify their skills through their evaluation error rate.

**Annotation Process.** Annotators verify and correct each example by consulting the original paper PDF. Table 7 in the Appendix showcases typical errors found by our annotators, sorted by frequency. The most common error is "Incorrect Win Condition", e.g., mistakenly marking the perplexity metric as higher is better. Additionally, the win conditions for certain metrics are context-dependent. For example, Attack Success Rate (ASR) should be maximized in adversarial attack papers but minimized in defense papers. Annotators fix these errors by rewriting examples or removing them when the results are fabricated or incomparable due to unfair experiment settings.

To incentivize our annotators to provide high-quality annotations, we design the following bonus schemes. Each annotation initially earns $5. However, if an annotation is found incorrect during cross-validation by another annotator, the original annotator gets a penalty of $3, while the annotator who catches the error gets an additional $3 reward. We conduct four rounds of cross-validation. In each round, we randomly sample 400 examples and send them to review by new annotators. From the first to the last round, the rate of identified mislabels decreased from 11% to 2.5%.

## 3 Building a System for Research Outcome Prediction

### 3.1 Retrieval

When predicting outcomes of new ideas, human researchers often draw inspiration from existing literature. While the exact same ideas do not exist, prior studies can offer indirect insights, transferable knowledge, or analysis of similar sub-components. Therefore, we develop an agentic retrieval module that iteratively performs the following four steps: query generation, paper retrieval, paper summarization, relevance checking and filtering. The final retrieval results will be used as parts of inputs for prompting or fine-tuning.

**Step 1: Query Generation.** At iteration $t$, given a research goal, two research ideas, and previous queries and retrieval results, our system first determines whether sufficient information has been collected, thus allowing for early exit. If further research is required, the system prompts an LM to generate a new query distinct from previous queries. Importantly, since at least one idea in the given comparison pair is entirely novel, the system would not try to directly search for identical idea comparisons. Instead, the query generation employs two strategies: 1) retrieving ideas that offer indirect or transferable insights, and 2) decomposing the novel idea into sub-components and retrieving related literature for these sub-components.

**Step 2: Paper Retrieval.** We use `https://exa.ai` as the paper search engine. Unlike conventional keyword search, they support neural search that can directly take natural language queries as inputs, e.g., "effectiveness of confidence calibration and iterative refinement in question answering". EXA also supports automatic query optimization [12] for better retrieval performance. For each query, we retrieve

Table 3: Impacts of paper summarization methods: reuse abstracts or summarize the full paper. Evaluated on 600 test examples.

| Method | Accuracy |
|---|---|
| GPT-4.1 | 42.0 |
| w/ RAG (abstract-only) | 38.8 |
| w/ RAG (whole paper) | 53.0 |

the top-15 relevant papers from arxiv.org. To prevent
information leakage from retrieval, we only retrieve papers published before July 1st, 2024.

**Step 3: Paper Summarization.** Prior work mainly resues paper abstracts as summaries [14, 9]. However, abstracts are often too brief to cover rich details (e.g., ablation studies or comparison with specific baselines). Therefore, we download each paper's PDF, and prompt LMs to summarize the paper with respect to the current query. As shown in Table 3, this substantially boosts the accuracy from 38.8% to 53.0%. We find that using abstracts alone biases the model towards favoring old ideas, while summarizing the whole paper alleviates such biases.

**Step 4: Relevance Checking and Filtering.** The initially retrieved 15 papers per query ensure coverage, but may also introduce irrelevant papers that may negatively mislead LMs' predictions. To alleviate this, we prompt LMs to judge the relevance of each paper summary (binary classification: relevant or irrelevant), and subsequently filter out those irrelevant ones.

### 3.2 Fine-tuning

Next, we fine-tune LMs to reason over the research goal, two research ideas, and all retrieval results, to make the final prediction. A straightforward setting is fine-tuning LMs with golden outcome labels.

In addition, we also explore fine-tuning LMs to generate chain-of-thought (CoT) reasoning [15] before the final prediction. High-quality CoTs are not available for this task. Thus, following recent practices [17], we use the LM to augment the training data with its own generated CoTs. Specifically, we sample multiple CoTs for each query, filter out CoTs that lead to incorrect predictions, and try several strategies to select one CoT from all candidates (e.g., random or LM-based selection).

## 4 Evaluation on Human-written Ideas

### 4.1 Experiment Setup

We use GPT-4.1 as the main base model. Additionally, we evaluate multiple frontier LMs such as o3 and Claude 3.5 Sonnet. The knowledge cut-off of all these tested models is before July 1st, 2024[1].

Prior work shows that LMs often yield inconsistent predictions when swapping orders of inputs in pair-wise comparison tasks [3]. To mitigate potential order bias, we obtain two predictions for each test example by swapping the order of two ideas. We only consider the predictions valid when both predictions are consistent; otherwise, inconsistent predictions are considered incorrect by default.

### 4.2 Automatic Evaluation

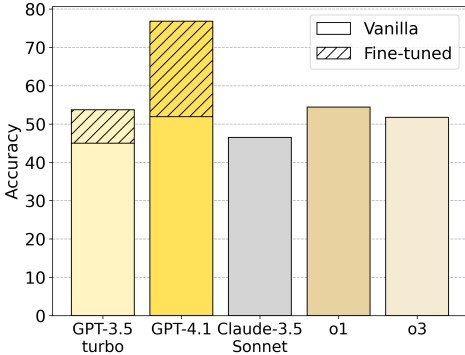

Figure 3: Automatic evaluation results.

Table 4: Fine-tuning GPT-4.1 with raw outcome labels or self-augmented CoTs.

| Model | Accuracy |
|---|---|
| GPT-4.1 | 51.4 |
| + FT on Raw Label | 77.0 |
| + FT on Random CoT | 48.0 |
| + FT on LM-selected CoT | 47.8 |

**Finding 1: LMs are not naturally good at the task.** We start with evaluating the zero-shot performance as a baseline. We select the best-performing zero-shot prompt among 10 candidates. As shown in Figure 3, even with the same retrieval augmentation, none of the frontier models are

---

[1]We didn't evaluate Claude 3.7 Sonnet since its knowledge cut-off is November 1st, 2024.

naturally good at this task. For example, GPT-4.1 only achieves an accuracy of 51.9%; reasoning models like o1 and o3 also just achieve similar or slightly better performance.

**Finding 2: LMs can learn to predict idea outcome via fine-tuning on past ideas.** Fine-tuning GPT-4.1 on past ideas substantially boosts its accuracy in predicting "future" ideas, from 51.9% to 77%. This demonstrates the importance of capability elicitation on this task.

However, fine-tuning GPT-3.5-turbo on the same data only yields a moderate improvement (45.0% to 53.7%). Given that GPT-3.5-turbo has a much earlier cut-off (September 1st, 2021), we suspect that GPT-4.1's performance arises from both its stronger general reasoning abilities and the updated knowledge of recent AI research — a crucial advantage given the rapid pace of AI in recent years.

In addition, the limited gains with GPT-3.5-turbo, a weaker but still capable model, reveal that our dataset does not have some trivial shortcuts that can be easily fit.

**Finding 3: Self-augmented CoTs do not improve performance.** Prior work [17] shows that fine-tuning LMs on their own generated CoTs can improve performance, e.g., in mathematical reasoning. On our benchmark, however, this approach yields no gain. As Table 4 shows, neither randomly sampled nor LM-selected CoTs improves upon the baseline. We suspect that because of GPT-4.1's near-chance performance, its generated CoTs are often not sound. Consequently, these low-quality CoTs provide little useful signals for fine-tuning.

### 4.3 Comparison with Expert Researchers

**Data Preparation.** Since our recruited annotators are primarily NLP researchers, we select five popular NLP topics: long-context, alignment, reasoning, agent, and efficient LM. For each topic, we sample 6 to 12 idea pairs from our test set, resulting in a total of 45 NLP idea pairs.

**Human Annotation.** For each topic, we recruit 5 researchers with relevant backgrounds. Among them, 68% have published papers on their assigned topics, while the remaining 32% also have extensive paper reading experience. Table 5 summarizes their profiles.

Given a test example, annotators first indicate whether they have read any of these two ideas. Then, they provide a binary prediction and a natural language rationale. The average time cost per annotation is 9.1 minutes. They are paid $50 for completing examples under their assigned topics. In total, we collect 225 labels; three are excluded since the annotators have read both ideas before. The final data thus has 222 valid annotations for the 45 idea pairs.

Table 5: Statistics of research profile and the time cost (minutes) per annotation of our recruited AI researchers.

| Metric | Mean | Min | Max |
|---|---|---|---|
| **Paper** | 18.6 | 2 | 90 |
| **Citation** | 1273.8 | 14 | 14,378 |
| **H-index** | 8.9 | 2 | 41 |
| **i10-index** | 9.3 | 1 | 56 |
| **Time Cost** | 9.1 | 4.5 | 15 |

**Results.** As shown in Figure 1 (left), even expert NLP researchers struggle to predict research outcomes in their own research domains. Specifically, doing majority voting over five experts' predictions on each example achieves only 48.9% accuracy. We further measure inter-annotator agreement, i.e., how often does each annotator's label agree with the majority voting label. The agreement score is 75.1%, which is similar to the inter-reviewer agreement on conference paper decisions, e.g., 75.8% in NeurIPS 2021 [11].

We also establish a ceiling human baseline by aggregating the best-performing researchers per research topic, where the best-performing researcher is selected based on the exact same test set. This ceiling human baseline yields a 60.0% accuracy. Overall, these results show that predicting an idea's chance of success is a difficult task that even domain experts struggle with. In comparison, our system achieves an accuracy of 64.4%, substantially surpassing human experts.

## 5 Stress-testing Our System's Robustness

We run extensive stress-testing to ensure that our system does not exploit ideas' superficial features, such as complexity and stylistic cues. Each test is based on one potential superficial feature that the model can potentially rely on. We divide our test set based on whether the label is correlated with the feature, e.g., examples where the more complex method performs better vs. examples

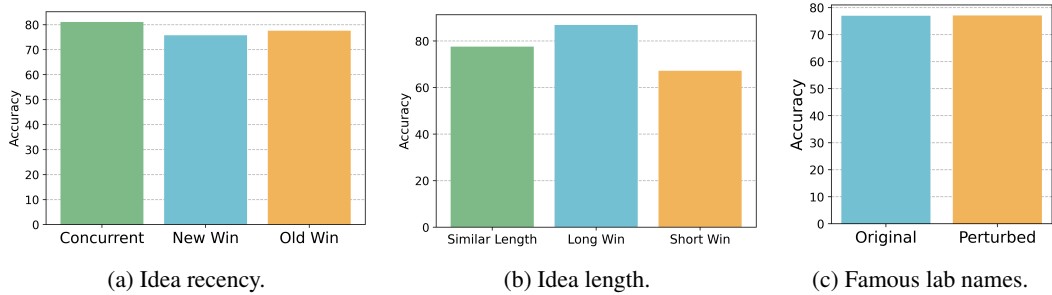

| (a) Idea recency. | (b) Idea length. | (c) Famous lab names. |

Figure 4: Stress-testing our system's sensitivity to three biases that humans might be prone to. The low accuracy variance across subsets of the test data indicates that our system is robust.

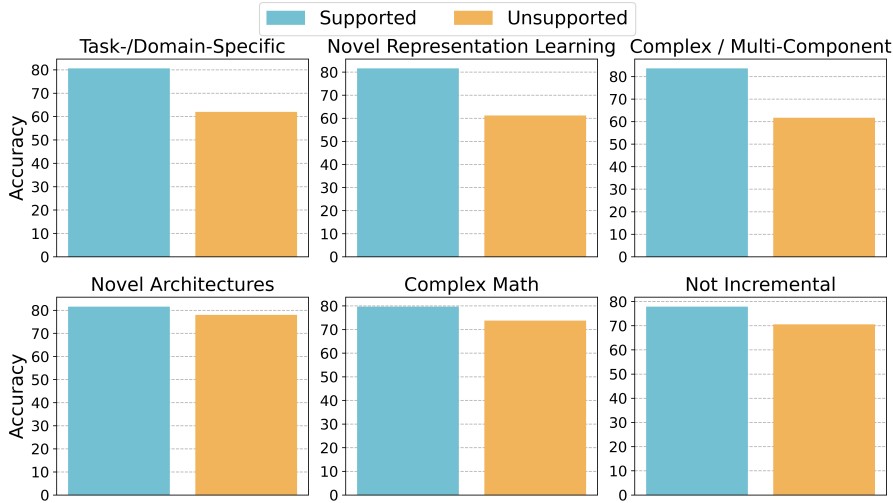

Figure 5: Our system is robust based on LM-designed stress tests. The *Supported* subset includes examples where the feature is noticeably more prominent in the winning idea, e.g., includes more complex math; the *Unsupported* subset is the opposite. Our system shows a slight (potentially justifiable) bias, but is generally robust.

where the simpler one is better. We then compare the model's accuracy on these subsets: a small accuracy difference indidates a robust model that's not overly sensitive to superficial features. We hand-designed three tests, then use an LM to propose many more tests.

## 5.1 Human-Designed Tests

We design three tests based on biases that human experts might be prone to when evaluating ideas.

**Recency**, i.e., favoring new ideas. We divide our test set into three sets based on whether the newer idea is better, or if the two ideas are concurrent (published within three months). Although we strip date information from idea descriptions, LMs might be able to infer that information based on prior knowledge and indirect cues such as references. As shown in Figure 4a, our system's accuracy is not sensitive to idea recency.

**Length**, i.e., favoring long and complicated ideas. We divide the test set into three based on whether the longer idea is better, and ties. Results shown in Figure 4b indicate that our system has a slight bias but can still identify complicated yet ineffective ideas.

**Famous Lab Names**, i.e., favoring ideas from "big names". To study this, we append the names of 10 famous labs (e.g., Anthropic, OpenAI, Google Deepmind) to the summary of the *losing* ideas. If our system exploits the shortcut, accuracy should notably drop after this perturbation. Results shown in Figure 4c indicate our system's resilience.

## 5.2 LM-Designed Tests

To improve the coverage of our stress-testing, we follow [18]and use LMs to automatically propose robustness tests at scale.

**Method.** Our pipeline consists of three steps: 1) proposing hypotheses about differences between two idea groups (model-preferred v.s. model-dispreferred), 2) verifying hypotheses across the test set, and 3) splitting the test set into supported and unsupported subsets according to each hypothesis.

In step 1, due to the limited model context length, each time we randomly sample 5 idea pairs and group them into model-preferred or model-dispreferred sets. We then query the LM: "how is the idea in Group A different from the idea in Group B". We repeat this 500 times to obtain 500 hypotheses. An example of LM-proposed hypothesis is: "Each research idea in Group A is more focused on utilizing advanced probabilistic and mathematical techniques for optimization and learning in machine learning models compared to Group B."

Since hypotheses are derived from only 5 examples, we further verify them across the whole test set. Specifically, for each example, we query the LM: "Compared to {losing idea}, is {winning idea} {Hypothesis}?". If so, this example is classified as supported under that hypothesis, because simply exploiting this shortcut can get perfect accuracy on such examples. Each hypothesis thus divides our test set into supported and unsupported subsets. For each hypothesis, we calculate a validity score, defined as the proportion of supported examples. We then discard hypotheses with scores outside the 25%-75% range, leaving us with 289 hypotheses.

For each test, the decrease in accuracy from the supported to the unsupported subset indicates sensitivity to the corresponding feature.

**Results.** Hypotheses are classified as *Flagged* if the accuracy on the corresponding unsupported subset drops below 62% (given the average accuracy of our system over the entire test set is 77%), otherwise they are marked as *Cleared*. Across all 289 verified hypotheses, our system achieves an average unsupported accuracy of 68.5%, and notably, over 88% (255 out of 289) are *Cleared*, indicating robustness against most hypothesized superficial biases.

Table 6: Results on LM-proposed hypothetical biases. Hypotheses are *Flagged* if the unsupported subset's accuracy falls below 62%, otherwise *Cleared*.

| Metric | All | Flagged | Cleared |
|---|---|---|---|
| Size | 289 | 34 | 255 |
| Acc (Supported) | 81.3 | 82.1 | 81.1 |
| Acc (Unsupported) | 68.5 | 61.3 | 69.4 |

Even among the 34 *Flagged* hypotheses, our model maintains a considerable average unsupported accuracy of 61.3%. The first row in Figure 5 shows the evaluation results of three flagged cases. In particular, we find two representative hypotheses: model-preferred ideas are often more complex or more specialized (i.e., task- or domain-specific). However, predicting more complicated or specialized yet ineffective ideas is inherently more challenging. Therefore, under such tests, our system's $> 61\%$ accuracy (Figure 5) is non-trivial.

Further, examining the 255 *Cleared* hypotheses reveals potential shortcuts that our model does not rely on. In the second row of Figure 5, we present the valuation results of three cleared cases, which stree-tests whether our system always prefers favor ideas that are 1) focused on novel architectures, 2) mathematically complex, or 3) entirely novel instead of doing incremental changes to existing methods. As we can see, our system remains robust under these features.

## 6 Evaluation on Novel Unpublished Ideas

So far, our evaluation is based on published ideas with control measures for contamination. The real test, however, is to make predictions about unpublished ideas. The bottleneck to such an evaluation is to obtain groundtruth by actually implementing the ideas.

We conduct a small-scale experiment with unpublished ideas with actual groundtruths. We start with a dataset of 35 ideas from an AI ideation study [14], half generated by their LM agent, and the other half generated by their recruited NLP experts on the spot. None of these ideas have been released anywhere on the internet. The research topics focus on novel prompting methods to improve various capabilities of LMs. The outcomes are obtained by expert researchers implementing the ideas, costing 103.4 hours per idea on average. For each idea, the experiments and results are documented as a

paper, from which we can extract the idea into our standard format for our benchmarking. We pair the proposed novel idea with its baseline and derive the golden label based on empirical outcomes. After excluding ties, we finally collect 33 labeled idea pairs.

Our system achieves an accuracy of 63.6%, demonstrating the potential to use our system as a reward model in a complete automated research pipeline, either to guide sampling at inference time, or optimizing the idea generation model using reinforcement learning.

# 7 Related Work

**Accelerating Research with AI.** With the overarching goal of using AI to build AI, recent work has explored integrating AI into the entire research workflow, including literature review [2], ideation [14, 4], experimental validation [6, 9], paper writing [9], and review [8]. Notably, one fully AI-generated paper passed the peer-review process of an ICLR workshop [16]. However, discovering such a workshop-level idea is non-trivial: Starting from 256 candidate papers, the authors manually select the top-3 papers since LM judges are often unreliable, while only one paper gets accepted. Importantly, executing 256 AI research ideas into whole papers is expensive and slow because of the significant GPU or API costs. To accelerate AI research, our paper aims to answer the question: "can we predict the outcomes of ideas before actually implementing them?"

**Research Evaluation.** Prior work mainly focuses on peer-review style evaluation; the goal is to predict scores that align with human reviewers based on fully written papers. However, such evaluations still require implementing ideas. Moreover, peer-review scores are often subjective. Critiques of novelty, excitement, and paper presentation are often independent of the ideas' actual empirical effectiveness. Consequently, humans can get misled to prefer "fancy" (e.g., mathematically complex) yet ineffective ideas. Instead, this papers focuses on objective empirical effectiveness, for each we can reliably establish ground truth labels.

**Research Outcome Prediction.** Prior work also attempts to use AI for predicting empirical research outcomes. The most relevant work is [10], which uses AIs to predict neuroscience results. However, their actual experiment setup is entirely different from ours. Specifically, given a published paper abstract, they use LMs to alter descriptions about results while keeping the method and background unchanged. The goal is then to distinguish the original abstract from its altered counterpart. This task is easy, even a fine-tuned Llama-2-7B model achieves an 80%+ accuracy. In contrast, we study a more realistic and challenging scenario: given two independently written idea summaries that exclude any empirical results, predict which is more effective.

**Forecasting.** Our task is a type of forecasting, i.e., forecasting the outcomes of ideas before running actual experiments to get the outcome. Prior attempts on neural forecasting methods demonstrate the gains from retrieval and scale (e.g., model and data scale) [19, 13, 1, 7]. For example, [5] builds a human-level forecasting system by retrieval augmentation and fine-tuning on historical data. Inspired by these works, our work demonstrates the potential of building specialized LMs to surpass human experts in forecasting outcomes of AI research ideas.

# 8 Discussion

**Limitations.** Our current system functions as a black-box and is not conducive to human-AI cooperation beyond prioritizing research ideas based on model predictions. And despite our best attempts to verify its robustness, we cannot rule out the possible reliance on spurious features.

**Future Work.** Our fine-tuning method is straightforward but works well. Future work can explore advanced modeling methods, e.g., inference-time simulation of experiments. Our system can also be used as a reward model to improve automated ideation systems.

**Conclusion.** A crucial skill in empirical research is being able to predict which ideas are more likely to work out. Human experts develop this skill by reading papers and observing experiments, we show that LMs can do this more efficiently; our LM system outperforms human experts at making direct predictions about the outcome of AI research without running experiments. In addition to higher accuracy, our system is robust to biases like recency that humans are prone to. The system also generalizes to completely novel, unpublished ideas, including AI-generated ones, demonstrating the potential of further accelerations of AI research.

## 9    Acknowledgements

This work was supported by Eric and Wendy Schmidt and Open Philanthropy.

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

Table 7: Typical errors found in the human verification process sorted by frequency. Human annotators will rewrite or directly remove the incorrect data.

| Error Type | Example | Solution |
| --- | --- | --- |
| Incorrect Win Condition | The metric (e.g., perplexity) is the lower the better, but is extracted as the higher the better. | Rewrite |
| Incorrect Empirical Results | The numerical results are incorrect or hallucinated (i.e., do not exist in the paper). | Rewrite & Remove |
| Confusing Benchmark & Metric Description | The metric name is mistakenly extracted as the benchmark name. | Rewrite |
| Incomparable Ideas | One of the idea adopts an unfair experiment setup to serve as a ceiling or flooring baseline[2]. | Remove |

[15] Jason Wei, Xuezhi Wang, Dale Schuurmans, Maarten Bosma, Fei Xia, Ed Chi, Quoc V Le, Denny Zhou, et al. Chain-of-thought prompting elicits reasoning in large language models. *Advances in neural information processing systems*, 35:24824–24837, 2022.

[16] Yutaro Yamada, Robert Tjarko Lange, Cong Lu, Shengran Hu, Chris Lu, Jakob Foerster, Jeff Clune, and David Ha. The ai scientist-v2: Workshop-level automated scientific discovery via agentic tree search. *arXiv preprint arXiv:2504.08066*, 2025.

[17] Eric Zelikman, Yuhuai Wu, Jesse Mu, and Noah D Goodman. Star: Self-taught reasoner bootstrapping reasoning with reasoning. In *Proc. the 36th International Conference on Neural Information Processing Systems*, volume 1126, 2024.

[18] Ruiqi Zhong, Peter Zhang, Steve Li, Jinwoo Ahn, Dan Klein, and Jacob Steinhardt. Goal driven discovery of distributional differences via language descriptions. *Advances in Neural Information Processing Systems*, 36:40204–40237, 2023.

[19] Andy Zou, Tristan Xiao, Ryan Jia, Joe Kwon, Mantas Mazeika, Richard Li, Dawn Song, Jacob Steinhardt, Owain Evans, and Dan Hendrycks. Forecasting future world events with neural networks. *arXiv preprint arXiv:2206.15474*, 2022.

# Appendix

## A  Human Verification of Test Examples

Table 7 presents the typical errors found by our human annotators during the verification process.

## B  LM-Designed Tests

Table 8: Examples of LM-proposed hypotheses.

| Hypothesis | Acc (Pos.) | Acc (Neg.) |
|---|---|---|
| Each research idea in Group A is more focused on specific domains or task improvements with detailed mechanisms and methodologies tailored for unique problem sets. | 80.6 | 62.0 |
| Group A ideas focus more on novel conceptual frameworks and mechanisms for learning representations and transferring knowledge. | 81.6 | 61.2 |
| Group A ideas focus more on novel machine learning and AI model architectures while Group B ideas concentrate on practical methodologies and algorithmic improvements in specific domains. | 81.2 | 73.5 |
| Each research idea in Group A is more focused on utilizing advanced probabilistic and mathematical techniques for optimization and learning in machine learning models compared to Group B | 79.7 | 73.8 |
| Each research idea in Group A is more focused on specific application areas or tasks compared to Group B's broader methodological innovations. | 79.2 | 73.0 |
| Group A provides more detailed descriptions and technical depth about model architectures and implementations whereas Group B offers more concise summaries and emphasizes key features and differences from existing methods. | 77.9 | 70.6 |
| Each research idea in Group A is more focused on innovative techniques for enhancing AI model efficiency and effectiveness without extensive model training or fine-tuning. | 80.2 | 71.0 |
| Each research idea in Group A is more focused on innovative architectural and training strategies tailored to specific types of data or tasks compared to Group B's emphasis on optimizing existing models and frameworks for efficiency and performance. | 83.2 | 64.6 |

