# OpenReview forum: "Predicting Empirical AI Research Outcomes with Language Models"
_NeurIPS.cc/2025/Conference — NeurIPS 2025 poster_

### Official Review · Reviewer_SEun · 2025-06-12

**Clarity:** 2
**Significance:** 3
**Originality:** 3
**Rating:** 4
**Confidence:** 4

**Summary:**

This paper proposes a framework to predict the experimental performance of machine learning methods from descriptions of research ideas. The authors create a dataset that includes pairwise comparisons of performance on benchmarks, constructed using results presented in conference papers. Their experiments show that existing models, including GPT-4.1 and o3, cannot effectively predict experimental results even when provided with related papers. The result shows that GPT-4.1 fine-tuned on their dataset achieves better performance compared to human experts in the pairwise performance prediction task.

**Questions:**

Please let me know if there are any inaccuracies or misunderstandings in the points raised under Weaknesses.

**Ethical Concerns:**

["NO or VERY MINOR ethics concerns only"]

**Final Justification:**

My primary concerns regarding (1) data contamination and (2) a possible shortcut by the proposed method have been addressed in the rebuttal. Accordingly, I consider this paper to be above the threshold.

However, I still believe that the paper should provide more detailed information about the dataset and experimental settings. Although the authors mentioned that they plan to include additional details, the current lack of clarity remains the primary reason I am not assigning a higher score. I am also concerned about certain aspects of the dataset configuration, as noted in my response to the rebuttal.

**Limitations:**

Yes.

**Quality:**

2

**Strengths And Weaknesses:**

### Strengths

* This paper targets a crucial and useful research topic. Predicting the experimental performance of research ideas would be a crucial part of research automation.

* Novel dataset made with manual verification. The dataset that includes pairs of research ideas and experimental results is novel and useful. While the dataset is automatically created using LLMs, the quality is verified by manual annotation.

### Weaknesses

Although the idea is promising, the paper does not provide sufficient information to establish the reliability of its claims. Additionally, I have concerns regarding potential dataset contamination. I recommend a substantial revision before the work is considered for publication.

* Concern about data contamination

Lines 72 and 93 suggest that the authors use conference publication dates as the publication dates of papers. However, many papers are publicly available as preprints several months before their conference publication, and LLMs are often exposed to these preprints. Moreover, the paper retrieval module (Line 135) retrieves documents from arXiv, and the retrieved papers may contain information about the target ideas. To more reliably avoid data contamination, the authors should consider using the preprint publication dates rather than the conference publication dates.

In addition, according to [the official information from OpenAI](https://openai.com/index/gpt-4-1/), the knowledge cutoff date of GPT-4.1 is June 2024, though the exact date is not specified. To ensure a more reliable exclusion of training data, I believe it would be safer to include only papers published after July 2024.

* Concern about the shortcut in the proposed method

As stated in Line 130, some of the compared methods were proposed prior to the knowledge cutoff date and may be included in the retrieved documents. Since newer methods often outperform older ones, a trivial baseline would be to recommend methods not present in the retrieved documents. I am concerned that the proposed fine-tuned model may exploit this shortcut when making predictions.

Although I consider the experiments in Section 6 essential for addressing this concern, the results are not compared with baseline methods or expert human performance.

* Limited information about the dataset creation process, the dataset, and the results

The paper lacks detailed statistics, example prompts, samples from the datasets, and raw outputs from the evaluated methods. I recommend including raw examples from the datasets as well as unprocessed outputs from all evaluated methods.

* Too many missing references

There exist many previous studies in automated research, but this paper cites very few papers. Particularly, the first two paragraphs of the introduction and the second paragraph of the related work section do not cite any papers, and the total number of cited papers is only 19.

* Minor suggestions

  * I consider the majority voting over all benchmarks (Line 88) to be too sensitive to noise. For example, when Idea A outperforms Idea B on 3 out of 5 benchmarks, I am unsure whether Idea A is better than Idea B in a statistically significant manner. I recommend pairing ideas with a larger gap.

---

> ### Author Rebuttal · Authors · 2025-07-26
>
> Thanks for appreciating the strength of our paper! We will address each of your questions below, and are willing to expand or provide further responses if necessary.
>
> > Concern about data contamination. Lines 72 and 93 suggest that the authors use conference publication dates as the publication dates of papers.
>
> We will clarify this confusion in revision. “Publication date” here refers to the date that the paper first appeared on arxiv (which we extract from its arxiv ID), not the date that it’s published at the conference.
>
> > the knowledge cutoff date of GPT-4.1 is June 2024, though the exact date is not specified. To ensure a more reliable exclusion of training data, I believe it would be safer to include only papers published after July 2024.
>
> We pick a subset of our original test set that includes 1,444 ideas published after July 2024. As shown in the Table below, our main findings remain the same: GPT-4.1 achieves 77.3% accuracy (compared to 77.0% on the 1,585 ideas after June).
>
> | Model         | Accuracy |
> | ------------- | -------- |
> | GPT-3.5-turbo | 44.8     |
> | + FT          | 53.9     |
> | GPT-4.1       | 51.3     |
> | + FT          | 77.3     |
>
>
> > Concern about the shortcut in the proposed method: simply selecting newer methods, which  often outperform older ones. Although I consider the experiments in Section 6 essential for addressing this concern, the results are not compared with baseline methods or expert human performance.
>
> We agree that simply selecting newer ideas is a very likely shortcut. We are happy to include the following result in our revision. Here we report the best possible performance achievable by simply selecting newer ideas. Our fine-tuned GPT-4.1 substantially outperforms this baseline (77.3% v.s. 53.0%), suggesting that it’s not overly reliant on idea newness in its predictions.
>
> | Model         | Accuracy |
> | ------------- | -------- |
> | Fine-tuned GPT-4.1       | 77.3     |
> | Always selecting newer ideas |   53.0  |
>
>
> > The paper lacks detailed statistics, example prompts, samples from the datasets, and raw outputs from the evaluated methods.
>
> Currently, dataset statistics are presented  in Table 2, and an excerpt of dataset example in Figure 2. We will include more detailed dataset examples and example prompts (e.g. prompts for query generation, summarization in Section 3) in the final revision.
> The raw outputs of our fine-tuned models only include “Idea A or B is better”, while the prompted models also generate CoTs before making final predictions; we are happy to include CoT examples in the appendix although they are not used in our method.
>
> > Missing references
>
> We will add the following papers about peer-review style research evaluation in our final revision. As discussed in the second paragraph of related work, these papers are based on fully written papers, so still requiring implementing ideas. Moreover, these papers focus on critiques of idea novelty, excitement, and paper presentation, which are often subjective and independent of the ideas actual empirical effectiveness. In particular, a recent paper [5] shows that the ideas with higher LM-judged novelty scores in fact lead to worse empirical outcomes.
>
> [1]  The AI Scientist: Towards Fully Automated Open-Ended Scientific Discovery. arXiv 2024.
>
> [2] The AI Scientist-v2: Workshop-Level Automated Scientific Discovery via Agentic
> Tree Search. arXiv 2025
>
> [3] GRAPHEVAL: A Lightweight Graph-based LLM Framework For Idea Evaluation. ICLR 2025
>
> [4] Chain of Ideas: Revolutionizing Research Via Novel Idea Development with LLM Agents. ICLR2025
>
> [5] The Ideation–Execution Gap: Execution Outcomes of LLM-Generated versus Human Research Ideas. arXiv 2025.
>
> >  I recommend pairing ideas with a larger gap.
>
> We’d like to clarify that by “gap” the reviewer is referring to the gap in win rates of two ideas on the set of benchmarks, for example, by only labeling one idea as the winner if it beats the competitor on 80% of the benchmarks.
>
> We use different gaps to filter our test set, and report the data size and the performance of our fine-tuned GPT-4.1 in the table below. As expected, increasing the gap leads to higher accuracy, probably because it reduces label noise or picks easier idea pairs.
>
> | Gap             | \|Test Set\| | Accuracy |
> | --------------- | ------------ | -------- |
> | > 50% (default) | 1,585        | 77.0     |
> | > 60%           | 1,585        | 77.0     |
> | > 70%           | 1,439        | 78.7     |
> | > 80%           | 1,281        | 80.8     |
> | > 90%           | 1,190        | 81.2     |
>
>
> We agree that the gap in determining labels is an important hyperparameter. In our data collection and training pipeline, that hyperparameter can be easily adjusted. We will clarify in our revision that we don’t impose any choice of this hyperparameter.

---

> > ### Comment · Reviewer_SEun · 2025-08-02
> > **Re: Rebuttal by Authors**
> >
> > Thank you for the rebuttal. My primary concern regarding data contamination has been resolved, and the additional results were also helpful. I will increase my score to 4.
> >
> > However, I remain highly concerned about the lack of important details. The claim that LLMs can predict the outcomes of AI research is quite strong, and therefore it is essential to provide more comprehensive information about the dataset and experimental settings in order to make the paper more credible and reliable.
> >
> > > “Publication date” here refers to the date that the paper first appeared on arxiv
> >
> > Thank you for the clarification!
> >
> > > We pick a subset of our original test set that includes 1,444 ideas published after July 2024
> >
> > Thank you for the additional result. However, could you clarify why you maintain the claim that "GPT-4.1’s cut-off date is June 1st, 2024" (Line 94) and continue to use the original dataset? I did not assert that this discrepancy would have a major impact on performance. Nonetheless, if the cut-off date is not actually June 1st, then the dataset is conceptually inconsistent with the stated experimental assumptions, which raises concerns about the validity of the evaluation.
> >
> > > Our fine-tuned GPT-4.1 substantially outperforms this baseline (77.3% v.s. 53.0%)
> >
> > Thank you for the additional result!
> >
> > > We will include more detailed dataset examples and example prompts
> >
> > Thank you! I believe that the current statistics and examples are far from sufficient to make the paper reliable and credible.
> >
> > > We’d like to clarify that by “gap” the reviewer is referring to the gap in win rates
> >
> > For clarification, am I correct in understanding that "> 50% (default)" means a method is considered better if it outperforms on more than 50% of benchmarks? If so, do you expect that LLMs should be able to identify a method as better even when it only shows an advantage on, for example, 51% of benchmarks? While I agree that the threshold is a tunable hyperparameter, I do not consider that the current default of 50% is a reasonable choice.

---

> ### Author Response · Authors · 2025-08-02
>
> > could you clarify why you maintain the claim that "GPT-4.1’s cut-off date is June 1st, 2024" (Line 94)
>
> We thought it was June 1st. We will update the main results with July 1st, 2024 as the cut-off date!
>
> > While I agree that the threshold is a tunable hyperparameter, I do not consider that the current default of 50% is a reasonable choice.
>
> We will add the results with different thresholds in the paper and emphasize that future work should tune this parameter based on their specific scenarios.

---

### Official Review · Reviewer_pUcN · 2025-06-17

**Clarity:** 2
**Significance:** 4
**Originality:** 4
**Rating:** 4
**Confidence:** 2

**Summary:**

This paper is about predicting which of two research ideas is more promising. To realize this, the authors collect a dataset and evaluate LLMs, including a novel AI based ideation agent. Results show that the best models compare favorably to humans.

**Questions:**

1) In Section 3, how does the system determine whether sufficient information has been collected?

2) In Section 3 Step 1, what is a query?

3) In l. 130, why is at least one idea completely novel?

4) What is the role of paper summarization in Section 3?

5) Why are the four steps necessary in Section 3 and what is their main purpose?

**Ethical Concerns:**

["NO or VERY MINOR ethics concerns only"]

**Final Justification:**

The authors and other reviews motivated me to increase my score, even though I continue not to be an expert for this paper

**Limitations:**

The limitation section is extremely brief. One could mention for example, that evaluating the outcome of new ideas could drastically increase human reviewing burdens, among many other.

**Paper Formatting Concerns:**

Seems good

**Quality:**

3

**Strengths And Weaknesses:**

**Strengths**
This could be a potentially very impactful paper, as the results are quite convincing, and the idea and task seem very novel. The paper is also thorough and presents several analyses.

**Weaknesses**
My main concern is that I do not understand the core Section 3. For example, what is happening in query generation? An example would really be extremely helpful and much appreciated.

---

> ### Author Rebuttal · Authors · 2025-07-26
>
> Thanks for appreciating the strength of our paper! We will address each of your questions below, and are willing to expand or provide further responses if necessary.
>
>
> > In Section 3, how does the system determine whether sufficient information has been collected?
>
> We prompt LMs to automatically determine this.
>
> System prompt:
> ```
> You are an expert researcher in comparing and predicting other proposed research ideas.
> You will be given access to a paper retrieval API, which you may use to survey the literature and find relevant papers to help you make your decision.
>
> You will be given 5 rounds to decide on which idea would work better, but you do not need to use them all.
> At any round, you may exit early and decide on the comparison result.
> ```
>
> Excerpt of Prompt
> ```
> First, determine if you need to search for relevant research papers to make an informed assessment
> ```
>
> > In Section 3 Step 1, what is a query?
>
> It’s the query sent to the paper retrieval engine.
> Specifically, we prompt LMs as follows:
> ```
> First, determine if you need to search for relevant research papers to make an informed assessment (e.g., check if there are similar ideas or supporting empirical evidence). If yes, provide a search query in this format:
> <search>
> your query (a natural language sentence, e.g., effectiveness of weight decay in noisy label learning, effectiveness of diffusion models in image style transfer compared to vision transformer, effectiveness of model editing in sequential editing setting)
> </search>
> Guideline for search queries:
> - At least one of the ideas is entirely new. This means that you won't find exact comparisons in the literature. Instead, search for ideas that could provide indirect or transferable insights, or you can break the new idea into smaller sub-ideas to identify related work for each sub-component.
> - Make queries as precise and context-specific as possible to maximize the relevance of the retrieved information.
> - Avoid redundancy:
>    - Do not repeat previously made queries.
>    - Avoid gathering excessive evidence about components you already understand well. Instead, prioritize components that need further investigation.
> ```
>
>
> Here are two examples of generated queries:
> - ```effectiveness of state space models v.s. GRU architectures in sequence modeling tasks```
> - ```effectiveness of code-based prompting v.s. Chain of thought prompting for logical reasoning tasks```
>
> > In l. 130, why is at least one idea completely novel?
>
> Because each test example contains at least one idea published after June 1st, 2024, the knowledge cut-off date of our base model GPT 4.1 (Line 93-95).
>
> > What is the role of paper summarization in Section 3?
>
> Adding the whole content of all retrieved relevant papers would 1) exceed the model context limits, and 2) introduce too much irrelevant, non-informative information that might negatively mislead LMs’ predictions (e.g. experiment setup, hyperparameter details, irrelevant results).
>
> > Why are the four steps necessary in Section 3 and what is their main purpose?
>
> As explained in Line 122-126, similar to how human researchers draw inspiration from existing literature, we empower LMs with this agentic retrieval engine to improve its predicting ability.
>
> The four steps in Section 3.1 are standard practice for building retrieval-augmented systems [1] [2] [3]. In particular, our paper summarization step is quite different: while prior work directly reuses paper abstracts as summaries, we download each paper’s PDF and prompt LMs to summarize the whole paper with respect to the current query. As shown in Table 3, this substantially boosts the accuracy from 38.8% to 53.0%.
>
> [1] The AI Scientist: Towards Fully Automated Open-Ended Scientific Discovery. arXiv 2024.
>
> [2] The AI Scientist-v2: Workshop-Level Automated Scientific Discovery via Agentic Tree Search. arXiv 2025
>
> [3] Can Language Models Generate Novel Research Ideas. ICLR 2025
>
> > The limitation section is extremely brief. One could mention for example, that evaluating the outcome of new ideas could drastically increase human reviewing burdens, among many other.
>
> We are happy to include more discussion on the social impact of our work and AI-automated research in general (e.g., the potential increase in human reviewing burdens, as the reviewer suggested) in camera ready with the extra space. Our current discussion focuses on the technical limitations of our approach.

---

> > ### Comment · Reviewer_pUcN · 2025-08-04
> >
> > Thanks to the authors. Your rebuttal has clarified some points for me. I will not update my score, but the overall rating for the paper is already quite good.

---

### Official Review · Reviewer_2pTv · 2025-06-30

**Clarity:** 3
**Significance:** 3
**Originality:** 3
**Rating:** 5
**Confidence:** 4

**Summary:**

The authors create a new benchmark to compare LLMs and human experts in predicting which of two research ideas will succeed. The benchmark is composed by scraping ideas and experimental results from conference papers, resulting in 1585 human-verified idea pairs published after the base model's cut-off date, and 6000 pairs for training. Four rounds of students checks every test pair, and a panel of 25 senior researchers supplies the human baseline. Off-the-shelf o3 and Claude 3.5 land at roughly 55% accuracy, just above the experts. Fine-tuning GPT-4.1 on the 6 000 training pairs and letting it read brief paper summaries through RAG results in 77%.

**Questions:**

- "We only consider the predictions valid when both predictions are consistent; otherwise, inconsistent predictions are considered incorrect by default." --> Could the authors indicate how many times this happens?
- The authors state in footnote 1 that Claude 3.7 is not evaluated, but mention Claude 3.7 in the introduction. I believe this should be Claude 3.5?

**Ethical Concerns:**

["NO or VERY MINOR ethics concerns only"]

**Final Justification:**

The rebuttal did not change my acceptance recommendation.

**Limitations:**

The authors indicate some limitations. However, I believe they could add that reproducibility is hard since a closed-source model has been fine-tuned, which is not publicly accessible. Also, the data is not shared, but the checklist indicates they will release it.

**Paper Formatting Concerns:**

Respects formatting.

**Quality:**

3

**Strengths And Weaknesses:**

Strenghts:
- The authors introduce a new strongly curated benchmark which measures the ability of LLMs to assess the potential success of research ideas.
- The authors show that fine-tuning GPT-4.1 outperforms other SOTA LLMs and verify the robustness of these results through many stress-testing studies (including evaluation on novel unpublished ideas).

Weaknesses:
- While it is a first attempt at such a dataset, the limitation to binary choice is quite limiting to real-life use cases.

---

> ### Author Rebuttal · Authors · 2025-07-26
>
> Thanks for appreciating the strength of our paper! We will address each of your questions below. Happy to expand or provide further responses if necessary.
>
> > "We only consider the predictions valid when both predictions are consistent; otherwise, inconsistent predictions are considered incorrect by default." --> Could the authors indicate how many times this happens?
>
> The table below reports the frequency of inconsistent predictions (i.e. inconsistency score).
> Our choice to consider inconsistent predictions as incorrect is in line with prior work ([1][2]) in pair-wise evaluation .
>
> | Model         | Inconsistency|
> | ------------- | ------------------ |
> | GPT-3.5-turbo | 14.1               |
> | + FT          | 20.8               |
> | GPT-4.1       | 16.3               |
> | + FT          | 11.4               |
>
> [1]  AlpacaEval : An Automatic Evaluator for Instruction-following Language Models. github
>
> [2] Length-Controlled AlpacaEval: A Simple Way to Debias Automatic Evaluators. arXiv 2024
>
>
> > The authors state in footnote 1 that Claude 3.7 is not evaluated, but mention Claude 3.7 in the introduction. I believe this should be Claude 3.5?
>
> Yes, it should be Claude 3.5. Thanks for pointing out this typo!
>
> > While it is a first attempt at such a dataset, the limitation to binary choice is quite limiting to real-life use cases.
>
> We actually think pair-wise comparison is well aligned with real-life use cases: 1) human researchers often seek to know whether a new idea is better than specific baselines, 2) in an automated research pipeline, a pair-wise reward model can be used to rerank model-generated ideas or directly optimize the idea generation model.

---

### Official Review · Reviewer_pjfg · 2025-07-02

**Clarity:** 3
**Significance:** 3
**Originality:** 3
**Rating:** 5
**Confidence:** 4

**Summary:**

The paper studies the topic of predicting outcomes of AI research ideas. A benchmark is proposed to faithfully evaluate the soundness of the AI-generated idea. The construction of the benchmark requires laborious efforts, including selecting papers, extracting ideas. The authors also propose a fine-tuned GPT-4.1 with a paper retrieval agent for this task. The proposed method outperforms existing methods.

**Questions:**

1. What are the retrieved papers used for in the final proposed system? Section 3.1 only mentions how to retrieve papers from the database, but it does not mention how retrieval results are used. Are these results simply used as parts of inputs, or are some specific modules proposed for processing this information?
2. Will the proposed benchmark be released upon the acceptance of the paper? The benchmark is one of the core contributions of the paper, so it is important to have the benchmark publicly available to the community.

**Ethical Concerns:**

["NO or VERY MINOR ethics concerns only"]

**Final Justification:**

I will keep my rating to accept the paper.

**Limitations:**

Yes

**Quality:**

3

**Strengths And Weaknesses:**

Strengths:
1. It is worthwhile to construct a benchmark to evaluate the soundness of the generated ideas in the domain of AI research.
2. It is laborious to construct such a benchmark, and the authors provided detailed construction procedures in the paper and it looks reasonable.
3. The paper proposes a framework to predict the outcomes, and the performance surpasses existing methods.
4. The paper includes extensive and robust experiments to verify the performance of the proposed framework, including an ablation study on retrieved information, CoT, and stress-testing etc.

Weaknesses:
1. What are the retrieved papers used for in the final proposed system? Section 3.1 only mentions how to retrieve papers from the database, but it does not mention how retrieval results are used. Are these results simply used as parts of inputs, or are some specific modules proposed for processing this information?
2. Will the proposed benchmark be released upon the acceptance of the paper? The benchmark is one of the core contributions of the paper, so it is important to have the benchmark publicly available to the community.

---

> ### Author Rebuttal · Authors · 2025-07-26
>
> Thanks for appreciating the strength of our paper! We will address each of your questions below, and are willing to expand or provide further responses if necessary.
>
> > What are the retrieved papers used for in the final proposed system? Section 3.1 only mentions how to retrieve papers from the database, but it does not mention how retrieval results are used. Are these results simply used as parts of inputs, or are some specific modules proposed for processing this information?
>
> Yes, the retrieved results are simply used as parts of inputs during prompting and fine-tuning.
>
> > Will the proposed benchmark be released upon the acceptance of the paper?
>
> Yes, we will release the benchmark.

---

### Note · Authors · 2025-08-12

We sincerely thank all ACs and reviewers for their feedback and engagement. We are pleased that all reviewers have provided positive assessments of our work, and consider all major concerns addressed.

To briefly summarize our key contributions: we create a benchmark for directly predicting the outcome of empirical AI research ideas, a challenging yet crucial task for accelerating AI research. We train a model for this task, and demonstrate through comprehensive evaluation that our model is more accurate than human experts on NLP research topics, does not exploit trivial features such as publication date, and is robust in the face of many distribution shifts.

We thank the reviewers for appreciating the strengths of our work, that we have created a high quality benchmark for a crucial capability (pjfg, 2pTv, pUcN, SEun) and that our model's performance is convincing (pjfg, 2pTv, pUcN). While reviewer SEun had initial concerns about the impact of two hyperparameters in the data curation process: cut-off date and threshold (when filtering low confidence idea pairs), we provided additional experiments demonstrating that our results stay consistent with other choices of these hyperparameters.

We deeply appreciate the rigorous and insightful review process, which helped improve our work. We will update our paper based on the feedback and incorporate additional results generated through the rebuttal process.

---

### Decision · Program_Chairs · 2025-09-17

**Decision:**

Accept (poster)

**Comment:**

This paper introduces a benchmark for predicting which research ideas will succeed, comparing LM performance with human experts. A retrieval-augmented GPT-4.1 system outperforms both experts and frontier LMs, with robustness checks and evaluations on unpublished ideas confirming its promise. Overall, it presents a novel and impactful direction for using LMs to accelerate AI research.
During the review process, all reviewers gave this paper a positive score. Meanwhile, they also pointed out some possible improvements, such as the risk of data contamination, missing references and data curation details, and lack of clarity in some sections. The authors resolved most of these concerns during the rebuttal.

By evaluating the novelty of research idea prediction, I recommend acceptance of this paper.